# Developmental differences in perceiving arousal and valence from dynamically unfolding emotional expressions

Nikol Tsenkova[1]*, Daniela Bahn[2], Christina Kauschke[2], Gudrun Schwarzer[1]

1 Psychology and Sports Science, Department of Developmental Psychology, Justus Liebig University, Giessen, Germany, 2 Institute of German Linguistics, Clinical Linguistics, Philipps University, Marburg, Germany

* nikol.tsenkova@psychol.uni-giessen.de

## Abstract

The development of emotion perception has predominantly been studied using static, unimodal stimuli featuring the faces of young adults. Most findings indicate a processing advantage for positive emotions in children (positivity bias) and a negativity bias in adults, although these results are usually task-dependent. We created a new stimulus database comprising digital avatars from four age groups, dynamically expressing happiness, happy-surprise, anger, and sadness in visual (face only) and visual-verbal (face and voice) conditions. To determine whether previously found biases would re-emerge with this new database, we tested the arousal and valence perception of positive and negative emotions in 6- and 7-year-old children and young adults. Our results revealed high correlations between children's and adults' responses but also significant differences: children rated negative expressions as more arousing compared to adults and positive emotions as more positive than adults. Additionally, visual-verbal presentations were perceived as more arousing than visual across both age groups. In terms of valence, all participants found positive emotions as more positive in the visual condition, whereas negative emotions were perceived as more negative in the visual-verbal condition. As one of the first studies to employ dynamically multimodal emotional expressions, our findings underscore the relevance of studying developmental differences in emotion perception using naturalistic stimuli.

## Introduction

Effectively perceiving emotions is crucial for successful interactions and cultivating lasting relationships [1]. Understanding the development of emotional perception is imperative, given its role in relationship quality [2], life satisfaction [3], and academic achievements [4] throughout the different stages of our lifespan. Consequently,

**Data availability statement:** Data are available from the OSF repository [URL: https://osf.io/3rd7m/]

**Funding:** This work was supported by the DFG (Deutsche Forschungsgemeinschaft) [grant number SFB/TRR 540 135/3 2014]. The funders had no role in study design, data collection and analysis, decision to publish, or preparation of the manuscript.

**Competing interests:** The authors have declared that no competing interests exist.

numerous studies have focused on exploring the developmental trajectory of emotion perception across various age groups – mostly in children [5,6] and adults [7], but also in adolescents [8,9], and older adults [10].

To examine how emotion perception develops over time, researchers employ various tools such as rating scales based on established affective models like the Russell's Circumplex Model of Affect [11]. Within it, emotions are plotted in a coordinate system based on their valence (x-axis) and arousal (y-axis) values. Valence denotes whether an emotional stimulus is perceived as positive/pleasant or negative/unpleasant, while arousal, ranging from high to low, can be conceptualized in two distinct ways. It can refer to an individual's subjective feeling about a stimulus (internal arousal) or to the perceived level of arousal of the stimulus itself, such as when viewing facial expressions of emotion (external arousal). Some studies indicate that children and adults rate emotions similarly in terms of valence and arousal [12,13], as shown by the high correlations between their responses. Although these rating studies reveal high correlations between both group's perceptions, some uncover notable age-dependent biases. For instance, Vesker and colleagues [14] had participants rate arousal and valence of positive and negative facial expressions using two Self-Assessment Manikin (SAM) scales [15]. Children rated positive expressions as significantly more arousing and positive as compared to adults, indicating a positivity bias. Similarly, in a subsequent study, Vesker et al. [16] discovered that children exhibit a positivity bias for valence by categorizing positive expressions faster and more accurately than negative ones. On the other hand, adults displayed the opposite bias, showing greater accuracy for negative expressions than for positive ones. Biases have also emerged when using different emotional modalities, for example from emotional words or body movements. For instance, 5- and 6-year-old children were found to be faster and more accurate in categorizing positive words compared to older children and adults [17]. Additionally, 5-year-old children were found to be more accurate with happy body movements, while adults with angry movements [18]. However, findings on positivity and negativity biases tend to be inconsistent, likely due to the specificity of the experimental tasks. While some tasks are more likely to produce a positivity bias (e.g., identification tasks: children [19], adults [20]; intensity and arousal ratings: children [21]), others have shown to induce a negativity bias (e.g., visual search tasks: both children and adults [22]; recognition tasks: younger and older adults [23]; for a comprehensive review on infants and children, see [24,25]).

Such inconsistencies highlight some limitations in traditional stimulus design, calling into question the ecological validity of these findings. Two recent meta-analyses have made similar observations in studies with young adults [26] and with children [27], underscoring the importance of employing multimodal dynamic stimuli to detect actual emotion perception age-dependent changes across the lifespan. In contrary, the majority of emotion perception studies rely on static images, such as still images of facial and bodily expressions [28]. Furthermore, emotion modalities are often studied in isolation from each other, which is rare in everyday life [29], and many existing emotion databases feature only high-intensity levels of emotion

expressions [30,31], which could be perceived as unnatural. Finally, most emotional stimuli databases lack the facial representation diversity encountered in everyday life and are typically expressed by young adults [32]. Several databases have addressed the issues of static presentation and isolated modalities – GEMEP [33], the CREMA-D [34], and the RAVDESS [35]. In all three, the highest emotion recognition accuracy was achieved in the multimodal audio-video condition (73%, 64%, and 80%, respectively) compared to the video-only (59%, 58%, and 75%) and the audio-only conditions (44%, 41%, and 60%). It appears that multimodal presentation enhances emotion recognition accuracy by at least 5% [36], while dynamic presentation, especially at low intensities, improves recognition compared to static presentation [37]. A handful of studies have dealt with the issue of age variation of presented faces, albeit using static stimuli like the Radboud Faces Database [38] which comprises photographs of emotional expressions from children and young adults. The participants' responses revealed that happiness was the most accurately recognized emotion (98% for adults' faces, 97% for children's faces). Zsido and colleagues [39] conducted a visual detection task using photographs of expressions displayed by child and adult models from various databases and further confirmed the happiness superiority effect across diverse age groups and stimuli. One study [40] using a dynamic database with unposed, spontaneous facial expressions of children showed once again that the highest accuracy came from identifying happiness (72%). Similarly, a recent study by Negrão et al. [41] utilizing dynamic presentations of children's facial expressions reaffirmed that happiness is the most easily recognized emotion, as agreed upon by four judges. We are aware of only one experiment [42] that explored how children and adults differed in labeling dynamic emotional expressions in three modalities separately (facial, vocal, and postural) and in a combined multimodal condition. Overall, children had the highest accuracy for sadness, followed by happiness and anger, with fear having the lowest recognition score. In contrast, adults were most accurate with happy expressions. Regarding the modalities, both age groups were most accurate with face-only and multimodal presentations, while the lowest accuracy came from the voice-only condition. This is consistent with previous research [43,44], which suggests that facial expressions alone are sufficient—or at least comparable—to bi- or multimodal information for emotion detection.

To date, and to the best of our knowledge, no study on the development of emotion perception has incorporated all relevant aspects of natural emotion perception: dynamic and simultaneous multimodal presentation (facial, postural, verbal information), lower emotion intensity, and diversity of facial age identities (childhood to late adulthood). A promising solution to address this gap involves utilizing technologies like the digital avatars called MetaHumans [45], which offer flexibility and realism in avatar creation and animation. MetaHumans offer several advantages in emotion research, including lifelike appearance, customizable characteristics (e.g., gender, age), precise expression intensity, and most importantly, full control and replicability of emotional expressions across different characters and conditions. Using MetaHumans allows for the creation of highly realistic digital characters while maintaining a great level of experimental control. However, the Uncanny Valley effect [46] poses a significant challenge when implementing digital avatars to study emotion perception [47]. This effect can produce feelings of discomfort and uneasiness upon encountering humanoid robots or digital humans. Despite this drawback, several studies featuring MetaHumans and other digital avatars have shown promising outcomes. Participants have reported higher levels of perceived attractiveness and lower levels of eeriness [48,49] and have experienced empathy when viewing emotional expressions [50]. Other studies have demonstrated the efficacy of digital avatars in eliciting emotional responses from participants despite their lower level of naturalness [51–53].

The current online rating study aims to investigate children's and adults' perceptions of valence and arousal in emotional expressions using stimuli that embody natural characteristics of emotions. To this end, we utilized a newly created stimulus database comprising digital avatars (MetaHumans), which features dynamic uni- and multimodal presentation, lower intensity, and a diverse range of facial ages. Specifically, we aim to explore the extent to which the previously reported positivity and negativity biases in the development of emotion perception are evident when using such natural stimulus material.

## Methods

### Participants

We tested two separate groups of German children, one for each of the two dependent measures (arousal and valence), and one group of adults who performed both measures simultaneously. In order to avoid fatigue and confusion regarding the difference between the scales, we separated the children sample into two groups. We did not expect such issues with the adult sample; hence, there was only one adult group.

The arousal group consisted of 26 children (age range: 6–7 years, $M_{age}$ = 6.4 years, $SD$ = 4 months, 16 females), and the valence group consisted of 28 children (age range: 6–7 years, $M_{age}$ = 6.4 years, $SD$ = 4 months, 13 females). The adult sample comprised 28 German young adults (age range: 19–35 years, $M_{age}$ = 25 years, $SD$ = 4.5 years, 16 females).

Children were recruited via the contact database of the Department of Developmental Psychology at the Justus Liebig University Giessen. This database consists of children of all ages (from infancy to adolescence), whose parents have given their consent to be contacted for experiments done in the Department of Developmental Psychology. The specific lists from which participants were contacted were children born in the months between May 2017 and February 2018, aged 6 and 7 years. For their participation, children received a 10-euro toyshop voucher. Adults were recruited via mailing lists of different universities (Giessen, Frankfurt, and Marburg) and were given student credits or a 15-euro digital voucher.

This study was approved by the local Ethics Committee of the Department of Psychology, Justus Liebig University of Giessen, Germany (#2021−0037). Consent for participation was obtained in written form before the beginning of the experiment—either from the adult participants themselves or, for the child participants, from their parents or caretakers.

We conducted an a priori G*Power analysis [54] to determine the required sample size for robust statistical power, with the criterion set at $\alpha$ = .05 and power = .95. A mixed analysis of variance (ANOVA) was selected as the statistical test, with two age groups as the between-subjects factor and four within-subjects measurements (positive visual, positive visual-verbal, negative visual, and negative visual-verbal). The results indicated that a minimum total sample size of 36 participants (18 per group) was sufficient to test the hypotheses of the current study.

### Stimuli

We created a new stimulus database of dynamic emotion expressions, incorporating both visual and verbal information. We named it Meta–MED (**Meta**Human **M**ultimodal **E**motion **D**ynamic database, link). For its creation, we used high-fidelity digital avatars called MetaHumans owned by Epic Games [45]. Our objectives during the creation process were the following:

I. Generating characters from four different age groups: young children (visually representing 6–7-year-olds), adolescents (visually representing 14–15-year-olds), young adults (visually representing individuals in their late 20s), and older adults (visually representing individuals in their 70s and above).

II. Including both female and male characters in each age group.

III. Creating two sets of visually distinct characters per age group and gender to investigate whether emotion perception is affected by the specific design of the characters.

IV. Producing dynamic, low–intensity animations for each character expressing four different emotions: two positive (happiness and happy-surprise) and two negative emotions (anger and sadness).

V. Developing two conditions for each emotion expression: one featuring only visual information (pure facial expression, from now on "visual condition"), and one combining facial and coinciding verbal information (from now on "visual-verbal condition").

### Creation of the Metahumans

To achieve objectives I, II, and III, we modified the appearance of existing character templates in the MetaHuman Creator (https://metahuman.unrealengine.com/), an online cloud-based tool for designing characters. We utilized the available templates for adolescents, young adults, and older adults. No templates were available for 6–7-year-old children, so we first created these using the MakeHuman software [55]. We then imported and stylized them in the MetaHuman Creator tool.

In total, we developed 16 characters, comprising two genders, four age groups, and two character versions, all of White origin. Finally, we animated these characters in Unreal Engine v. 5.1 [45].

### Dynamic presentation of the Metahumans

To achieve objectives IV and V—animating the four emotions and creating the visual and visual-verbal conditions—we used Face Control Rig within Unreal Engine. This in-built tool is based on the Facial Action Coding System (FACS) developed by Ekman and Friesen [56], which provides detailed information on the precise movements of facial muscles associated with different emotions. The Face Control Rig enables the creation of nuanced facial expressions, such as furrowing the brows to convey anger or raising the corners of the mouth to indicate happiness. Furthermore, to ensure a natural depiction of these emotions, we observed videos of actors showing the same facial expressions using FACS (ADFES database [57]) to fine-tune the animated expressions accordingly.

Additionally, we used the Face Control Rig to animate lip movements for the visual-verbal condition, where emotional expressions were synchronized with spoken sentences. We selected simple and clear sentences that align with the expressed emotions to ensure they are suitable for children as young as six years. The sentence structure across all emotions was: "Ich bin + specific word for emotion", resulting in the following sentences: "Ich bin glücklich" ("I am happy"), "Ich bin traurig" ("I am sad"), "Ich bin überrascht" ("I am surprised"), and "Ich bin wütend" ("I am angry"). All verbal information accompanies the visual expression, with the exception of happy-surprise, where the characters first display the emotion and then speak.

Once a certain condition was successfully animated for one MetaHuman (e.g., happiness in a child character), the facial movements were replicated across all other characters. Each emotion animation was limited to two seconds, reflecting the quick onset and brief duration typically associated with natural emotional expressions [58]. Research shows that shorter displays of emotional expressions generally yield higher accuracy in recognition [59] and are perceived as more realistic when lasting under 1,500 milliseconds [60]. Each emotional expression begins with a neutral face and gradually unfolds into the full emotion, maintaining a lower intensity to ensure a natural appearance [61,62].

For the vocalizations, we obtained the recordings from individuals in the same age groups as the characters in the stimuli (children, adolescents, young adults, and older adults). They were asked to record their voices at home with a phone recorder and were instructed to "feel the emotion" when pronouncing the sentences for a more realistic representation as opposed to neutral pronunciation. Their approval to use their voices for the stimuli was successfully received via consent forms.

The final dataset consist of 128 individual video clips (2 dataset versions x 8 characters x 4 emotions x 2 conditions). For static examples of the stimuli, see Fig 1. Dynamic examples are available at this link.

### Procedure

For the online rating study, we used the SoSci survey platform [63], which participants accessed from their personal computers. The presentation of stimuli was pseudo-randomized in four different sequences to avoid repeating the same emotion, character, or age group consecutively. Each of the four presentation versions started with a different emotion, age group, and character. Participants were randomly assigned to one of the four presentation sequences using the randomization function available on SoSci. An optional one-minute break was provided halfway through.

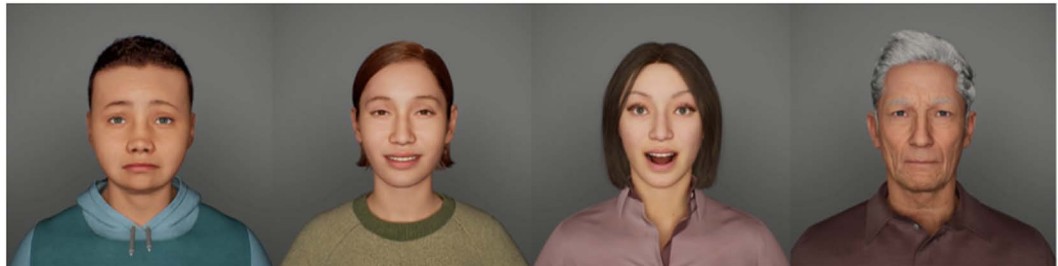

**Fig 1. Examples of the stimuli.** Still images of the digital avatars from the four age groups (from youngest to oldest) expressing all emotions (left to right: sadness, happiness, happy-surprise, anger). For the dynamic version, please refer to the link provided in the Methods/Stimuli section.

At the end of the experiment, we included three additional scales from the Godspeed Questionnaire [64] to test for Uncanny Valley effects related to the overall impression of the new stimulus dataset. The Godspeed Questionnaire (GQ) is the most frequently used tool in the field of Human-Robot Interaction [65], with over 3,000 citations as of April 2025, and has been translated into 19 languages. It consists of five scales, which can be used independently: Anthropomorphism ($\alpha = 0.87$), Animacy ($\alpha = 0.92$), Likeability ($\alpha = 0.70$), Perceived Intelligence ($\alpha = 0.75$), and Perceived Safety ($\alpha = 0.91$) [66]. The German version of the GQ [67] has been reported to have good internal reliability ($\alpha = 0.70$). In terms of validity, only the Polish version [68] has undergone factor analysis, which yielded a total variance of 74.24% for the four-factor solution.

We took two items from the scale Likeability and one item from the scale Anthropomorphism. We included a rating scale for each ranging from 1 to 5, with the midpoint (3) labeled as "I don't know". The first item from the scale Likeability (also called Likeability in our experiment) measured participants' perception of how nice or scary the MetaHumans appeared, ranging from "very scary" to "very nice". The second item from the scale Likeability (Friendliness) determined the extent to which the MetaHumans seemed friendly, with responses ranging from "very unfriendly" to "very friendly". The third item from the scale Anthropomorphism (Realism) evaluated how realistic the characters seemed, ranging from "very unrealistic" to "very realistic".

Participants rated arousal and valence perceived in the emotional expression videos by using two SAM scales [15]. The valence scale features figures with facial expressions ranging from frowning to smiling. The arousal scale includes figures depicting states from calm and relaxed to excited. Participants selected the figure that best represented the emotion in the video, rather than how they subjectively felt about it.

To achieve this in our children sample, parents received detailed instructions at the beginning of the experiment. They assisted their children to access the online study, understand the task, and complete the ratings. Parents were advised not to influence their children's responses to ensure the collection of their subjective perceptions. Children were randomly assigned to one of the two character database versions and completed 64 trials for either valence or arousal.

At the start of the experiment, participants completed two practice trials with characters not featured in the main dataset. Each experimental trial presented a video of an emotional expression on a single page, with the SAM scales and the following prompts displayed underneath:

 I. For valence: "Geben Sie bitte an, wie unangenehm/angenehm das gezeigte Gefühl wirkt." ("Please indicate how unpleasant/pleasant the shown feeling appears.").

 II. For arousal: "Geben Sie bitte an, wie ruhig/aufgeregt das gezeigte Gefühl wirkt." ("Please indicate how calm/excited the shown feeling appears."). For an example of a trial page and the SAM scales, see S1_Trial example.

Videos could be played only twice; after this, the play button was disabled and the videos faded to black to prevent replays. Selecting an answer was mandatory to proceed to the next page/trial.

Adults received a SoSci link including both arousal and valence scales shown simultaneously, one above the other. They completed both dataset versions for a total of 128 trials. The procedure and task were otherwise identical to those used for the children.

Responses were collected in the period between 01/06/2023 and 31/10/2023.

## Results

### Preliminary analyses

To test the normality of our arousal and valence data, we performed Shapiro–Wilk tests, which confirmed that the data met the assumptions for parametric analysis. Furthermore, we found no significant gender effects for either of the two measures – for arousal, $F(1, 50) = .11$, $p = .73$, $\eta2p = .002$, for valence, $F(1, 52) = .23$, $p = .63$, $\eta2p = .004$. Thus, we did not include gender in all further analyses. Correlational heatmaps can be found in S2_Correlational heatmaps.

### Effects of different stimuli versions

To investigate whether the visual appearance of the MetaHuman characters influenced the participants' responses, we compared the ratings given for the two datasets, each featuring distinct sets of MetaHuman characters.

For the child sample, given that each child saw only one version, we performed independent-sample $t$-tests. For arousal, we found no significant differences, $t(24) = 0.01$, $p = .98$. For valence, the results showed no significant difference between children's ratings of the two versions, $t(26) = 0.35$, $p = .72$.

For the adult sample, where all participants saw both versions, we used paired-sample t-tests. For arousal, we found no significant difference, $t(27) = -1.16$, $p = .25$, comparing the two versions.

A significant result emerged for valence, $t(27) = 2.30$, $p = .02$. To further investigate, we compared the positive and negative emotion ratings between version one and version two. The significant difference stemmed from the positive emotions, $t(27) = 2.40$, $p = .02$, with version one ($M_1 = 4.18$, $SD = 0.29$) being rated slightly higher in valence compared to version two ($M_2 = 4.13$, $SD = 0.28$). To identify which specific positive emotion (happiness or happy-surprise) in which stimulus condition (visual or visual-verbal) contributed to this difference, we performed four additional paired-sample t-tests. A significant difference was found in the visual-verbal happiness ratings, $t(27) = 2.51$, $p = .01$, with version one ($M_1 = 4.20$, $SD = 0.29$) being higher than version two ($M_2 = 4.08$, $SD = 0.34$). Given the high number of comparisons (6), we applied a Bonferroni correction to adjust the alpha level to 0.008. After correction, the initially significant difference in visual-verbal happiness was no longer significant..

Thus, while the visual appearance of the MetaHumans had a small influence on emotional responses in adults, the effects were not consistent across all measures. Therefore, for subsequent analyses, we used the mean ratings of the adults' responses from both versions.

### Main results: arousal

**Correlations between children and adults.** To explore whether children and adults' arousal responses were similar, we performed two Spearman correlations. These correlations were based on the averaged arousal ratings for each Stimulus Type (visual and visual-verbal) across the two groups. As seen in Fig 2 there is a high overlap between the two groups for the visual condition, which was confirmed by a strong correlation, $r_s(30) = .75$, $p < .001$.

Fig 3 also shows an overlap between the arousal ratings of both age groups in the visual-verbal condition supported by a significant correlation in the visual-verbal condition, $r_s(30) = 0.70$, $p < .001$. Overall, these findings suggest that both age groups had comparable arousal responses to both the visual and visual-verbal stimuli.

**Role of age group, age of faces, valence category, and stimulus type.** To get a better understanding of the factors that influenced participants' arousal ratings, we conducted a repeated measures ANOVA using SPSS version 28

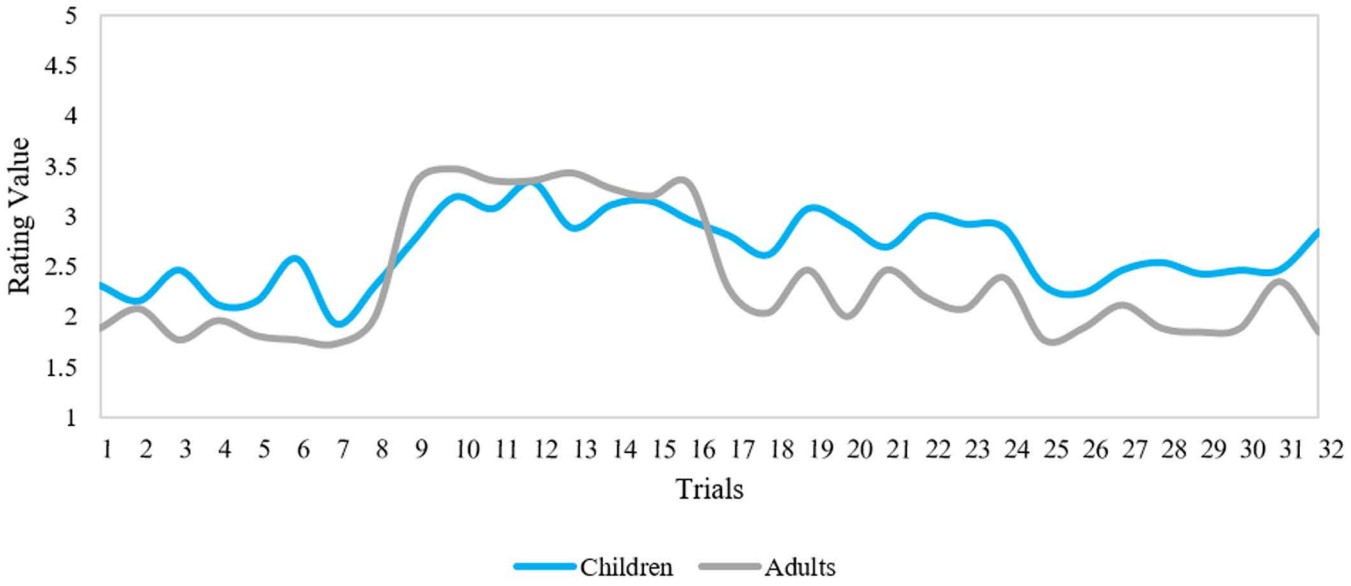

**Fig 2. Arousal ratings of both age groups across all stimuli in the visual condition.** The y-axis denotes the figure scale numerically (1 = low arousal; 5 = high arousal). The x-axis denotes all trials with 1-16 for positive expressions and 17-32 for negative expressions.

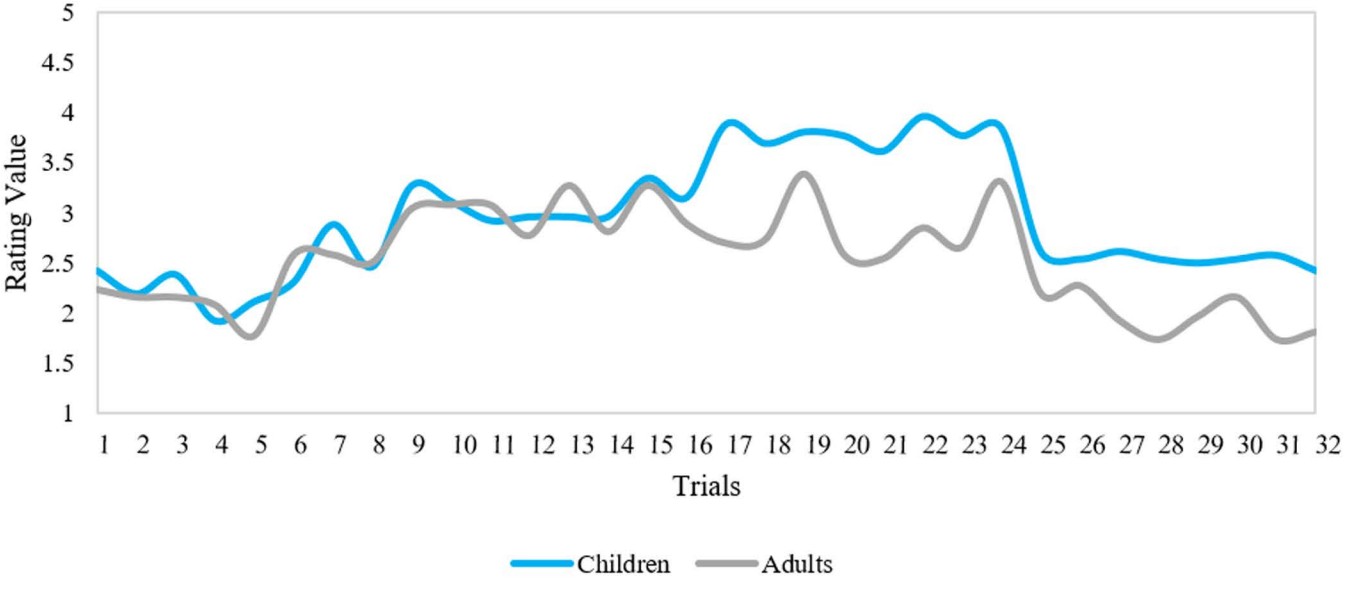

**Fig 3. Arousal ratings of both age groups across all stimuli in the visual-verbal condition.** The y-axis denotes the figure scale numerically (1 = low arousal; 5 = high arousal). The x-axis denotes all trials with 1-16 for positive expressions and 17-32 for negative expressions.

[69] examining the role of participants' age group, the age of the MetaHuman faces, the emotional valence of the facial expressions, and the type of stimulus presentation.

The analysis comprised four factors: (i) the participants' Age Group (children and adults), (ii) the Age of Faces of the digital characters (children, adolescents, young adults, older adults), (iii) the Valence Category (positive or negative),

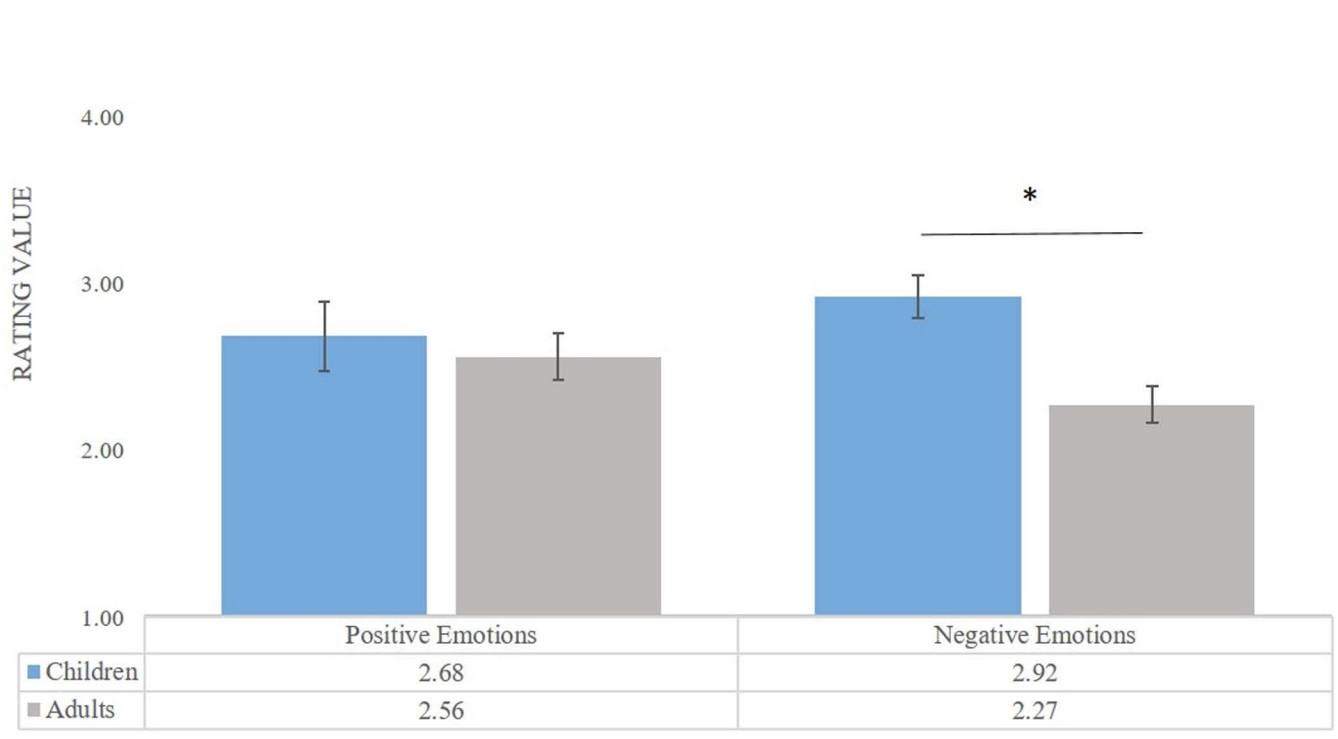

**Fig 4. Children's and adults' arousal ratings.** Mean rating values are for positive and negative expressions across both stimulus versions (visual and visual-verbal). Error bars represent standard error. *p < .05 **p < .01. ***p < .001.

and (iv) the Stimulus Type (visual or visual-verbal). Age Group served as a between-subjects factor, whereas the Age of Faces, Valence Category and Stimulus Type were treated as within-subject factors. The reported post-hoc analyses are Bonferroni-corrected pairwise comparisons.

A significant main effect of Age Group was found, $F(1, 52) = 5.55$, $p = .02$, $\eta^2 p = .096$, with children giving higher arousal ratings compared to adults ($M_{child} = 2.80$, $M_{adult} = 2.45$).

Another significant main effect was observed for the Age of Faces, $F(3, 156) = 2.60$, $p = .05$, $\eta^2 p = .05$, where faces of older adults rated as more arousing compared to faces of young adults ($M_{oa} = 2.68$, $M_{ya} 2.59$). However, this effect was not significant after Bonferroni correction ($p = .10$).

Furthermore, we found a significant main effect of Stimulus Type, $F(1, 52) = 17.25$, $p < .001$, $\eta^2 p = .25$. Post-hoc comparisons revealed that the visual-verbal condition had higher arousal ratings than the visual condition ($M_{vv} = 2.73$, $M_{v} = 2.51$, $p < .001$).

A significant interaction between Valence Category and Age Group emerged, $F(1, 52) = 3.87$, $p = .05$, $\eta^2 p = .07$ (see Fig 4). Post-hoc comparisons revealed that children rated negative emotions as more arousing than adults ($M_{child} = 2.92$, $M_{adult} = 2.27$, $p < .001$).

We observed a significant three-way interaction between Age of Faces, Valence Category, and Stimulus Type, $F(3,156) = 6.81$, $p < .001$, $\eta^2 p = .12$. First, faces of adolescents and older adults were rated as more arousing ($M_{ado} = 2.46$, $p = .02$; $M_{oa} = 2.49$, $p = .02$) than children's faces in the negative visual condition ($M_{child} = 2.25$). Second, in the positive visual-verbal condition, older adult faces were rated higher ($M_{oa} = 2.89$) than adolescent faces ($M_{ado} = 2.53$, $p < .001$) and young adult

faces ($M_{ya}$ = 2.59, $p$ = .004). Additionally, children faces were rated as more arousing in the positive visual condition than in the negative visual condition ($M_{posV}$ = 2.65, $M_{negV}$ = 2.25, $p$ = .01). Finally, faces of all ages were rated as more arousing in the negative visual-verbal than in the negative visual condition: children ($M_{negV}$ = 2.25, $M_{negVV}$ = 2.84, $p$ < .001), adolescents ($M_{negV}$ = 2.46, $M_{negVV}$ = 2.79, $p$ = .02), young adults ($M_{negV}$ = 2.39, $M_{negVV}$ = 2.76, $p$ < .001), and older adults ($M_{negV}$ = 2.49, $M_{negVV}$ = 2.77, $p$ = .003). Adolescent faces were rated as more arousing in the positive visual condition compared to the positive visual-verbal condition ($M_{posV}$ = 2.70, $M_{posVV}$ = 2.53, $p$ = .02), while older adult faces had higher arousal in the positive visual-verbal condition than in the positive visual condition ($M_{posV}$ = 2.58, $M_{posVV}$ = 2.89, $p$ = .001).

Thus, the findings show that children found emotional stimuli—specifically negative ones—as more arousing than adults. Additionally, visual-verbal stimuli were generally more arousing than visual-only stimuli, and character age significantly influenced responses depending on context: older adult faces elicited higher arousal than other face ages in the positive visual-verbal condition, while adolescent and older faces were more arousing than children's in the negative visual condition.

### Emotion-specific results

To determine which negative emotion elicited stronger arousal responses in children, we compared their ratings of anger and sadness. To do so, we conducted a paired-sample t-test comparing the two negative emotions. Results confirmed that anger elicited significantly higher arousal than sadness, ($M_{anger}$ = 3.33, $M_{sadness}$ = 2.50), $t(25)$ = 5.20, $p$ < .001.

### Observed power

We performed a post-hoc G-Power analysis for arousal (8 measurements, 54 participants) revealed observed power of 71.9%, slightly below the conventional threshold of .80.

### Main results: valence

To determine whether the valence responses of children and adults were similar, we performed two Spearman correlations. These correlations were based on the averaged valence ratings for each Stimulus Type (visual and visual-verbal) across the two groups. As shown by the overlap in Fig 5, the two age groups' responses were highly correlated, $r_s(30)$ = 0.97, $p$ < .001.

In the visual-verbal condition, there was a very strong correlation between the responses of the two groups, $r_s(30)$ = 0.94, $p$ < .001, as seen in the overlap in Fig 6. Similarly to the arousal results, these findings point to highly comparable valence responses of the two age groups to both the visual and visual-verbal stimuli.

### Role of age group, age of faces, valence category, and stimulus type

To grasp the influence of the different factors that influenced participants' valence ratings, we conducted a repeated measures ANOVA where we explored the role of participants' age group, the age of the MetaHuman faces, the emotional valence of the facial expressions, and the type of stimulus presentation. The reported post-hoc analyses are Bonferroni-corrected pairwise comparisons.

We found a significant main effect of Age Group, $F(1, 54)$ = 17.99, $p$ < .001, $\eta^2 p$ = .25, with children giving higher valence ratings compared to adults ($M_{child}$ = 3.32, $M_{adult}$ = 3.14).

There was also a significant main effect of Age of Faces, $F(3, 162)$ = 3.99, $p$ = .009, $\eta^2 p$ = .07. Faces of older adults received higher valence ratings compared to those of adolescents ($M_{ado}$ = 3.19, $M_{oa}$ = 3.27, $p$ = .01).

This finding is better understood with the significant interaction between Age of Faces and Valence Category, $F(3, 162)$ = 4.68, $p$ = .004, $\eta^2 p$ = .08, which revealed that this effect comes from positive emotions ($M_{ado}$ = 4.27, $M_{oa}$ = 4.43, $p$ < .001). Additionally, the post-hoc comparisons revealed that older adults' faces were perceived as more positive than those of children ($M_{child}$ = 4.33, $M_{oa}$ = 4.43, $p$ = .007).

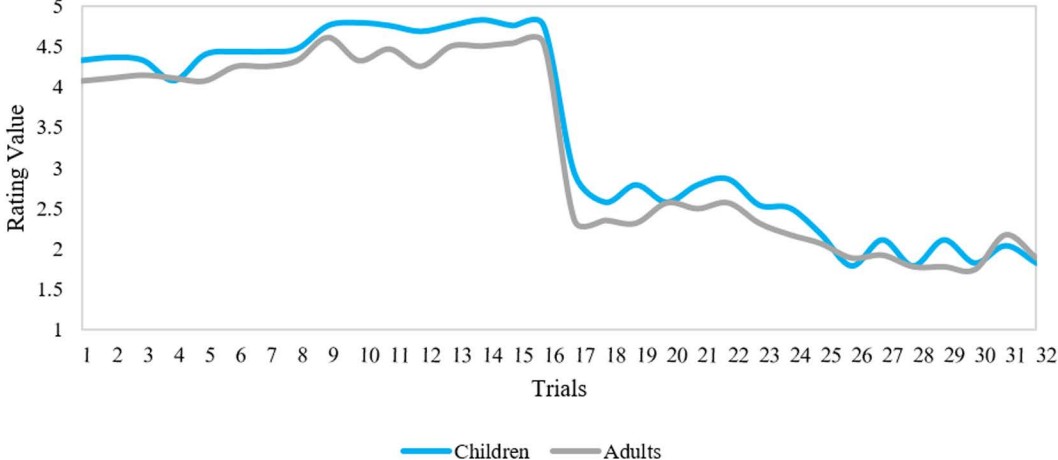

**Fig 5. Valence ratings of both age groups across all stimuli in the visual-verbal condition.** The y-axis denotes the figure scale numerically (1 = low valence; 5 = high valence). The x-axis denotes all trials with 1-16 for positive expressions and 17-32 for negative expressions.

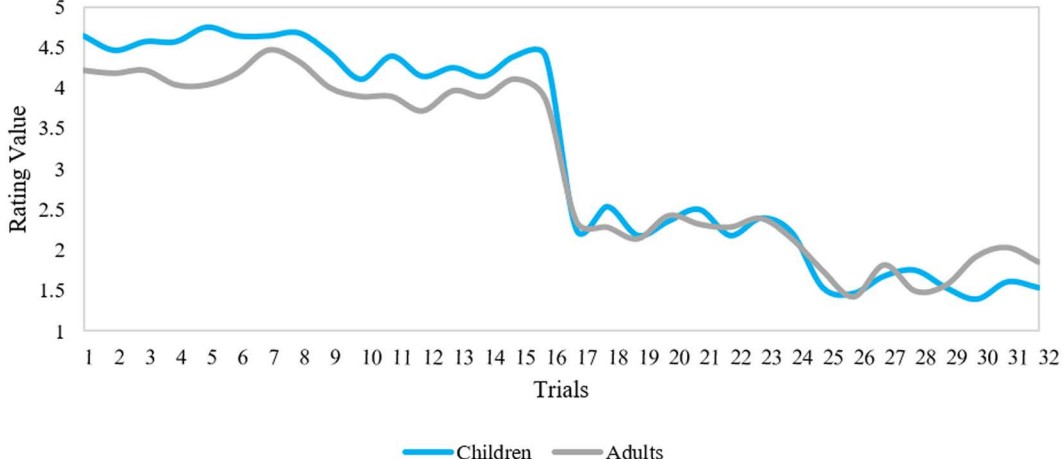

**Fig 6. Valence ratings of both age groups across all stimuli in the visual-verbal condition.** The y-axis denotes the figure scale numerically (1 = low valence; 5 = high valence). The x-axis denotes all trials with 1-16 for positive expressions and 17-32 for negative expressions.

A main effect of Valence Category emerged, $F(1, 54) = 1198.54$, $p < .001$, $\eta^2p = .96$, which is unsurprising considering that valence responses for negative and positive emotions should represent two extremes (i.e., low scores for negative emotions, high scores for positive).

In addition, there was a significant main effect of Stimulus Type, $F(1, 54) = 100.16$, $p < .001$, $\eta2p = .65$, showing that the visual condition was rated higher in valence as compared to the visual-verbal condition ($M_v = 3.34$, $M_{vv} = 3.12$, $p < .001$).

An interaction between Age of Faces and Stimulus Type was found, $F(1, 162) = 3.30$, $p = .02$, $\eta^2p = .06$. All faces were rated higher in valence in the visual compared to the visual-verbal condition: children ($M_v = 3.34$, $M_{vv} = 3.10$, $p < .001$), adolescents ($M_v = 3.30$, $M_{vv} = 3.09$, $p < .001$), young adults ($M_v = 3.37$, $M_{vv} = 3.10$, $p < .001$), and older adults ($M_v = 3.34$, $M_{vv} = 3.19$, $p < .001$).

Finally, we discovered a three-way interaction between Age Group, Valence Category, and Stimulus Type ($F(1, 54) =$ 12.66, $p < .001$, $\eta^2 p = .19$) as seen in Fig 7. Post-hoc pairwise comparisons revealed that children rated positive emotions as more positive compared to adults in both the visual ($M_{childV} = 4.55$, $M_{adultV} = 4.31$, $p = .004$) and visual-verbal conditions ($M_{childVV} = 4.45$, $M_{adultVV} = 4.10$, $p < .001$). Additionally, adults rated negative emotions as more negative compared to children only in the visual condition ($M_{adultV} = 2.15$, $M_{childV} = 2.32$, $p = .04$). Furthermore, children rated positive emotions as more positive in the visual condition compared to the visual-verbal ($M_{childVP} = 4.55$, $M_{childVVP} = 4.45$, $p = .01$), whereas negative emotions were perceived as more negative in the visual-verbal condition compared to the visual ($M_{childVVN} = 1.94$, $M_{childVN} = 2.32$, $p < .001$). Adult participants exhibited the same tendencies: positive emotions were rated higher in valence in the visual ($M_{adultVP} = 4.31$, $M_{adultVVP} = 4.10$, $p < .001$), while negative emotions were rated lower in valence in the negative visual-verbal condition ($M_{adultVVN} = 2.02$, $M_{adultVN} = 2.15$, $p = .02$).

Thus, the valence ratings revealed that children generally perceived positive facial expressions as more positive than adults. Visual stimuli produced stronger valence responses than visual-verbal ones, while faces of older adults received higher positivity ratings (in the visual condition). Furthermore, emotions were perceived as more positively in the visual than in the visual-verbal condition, and more negatively in the visual-verbal condition.

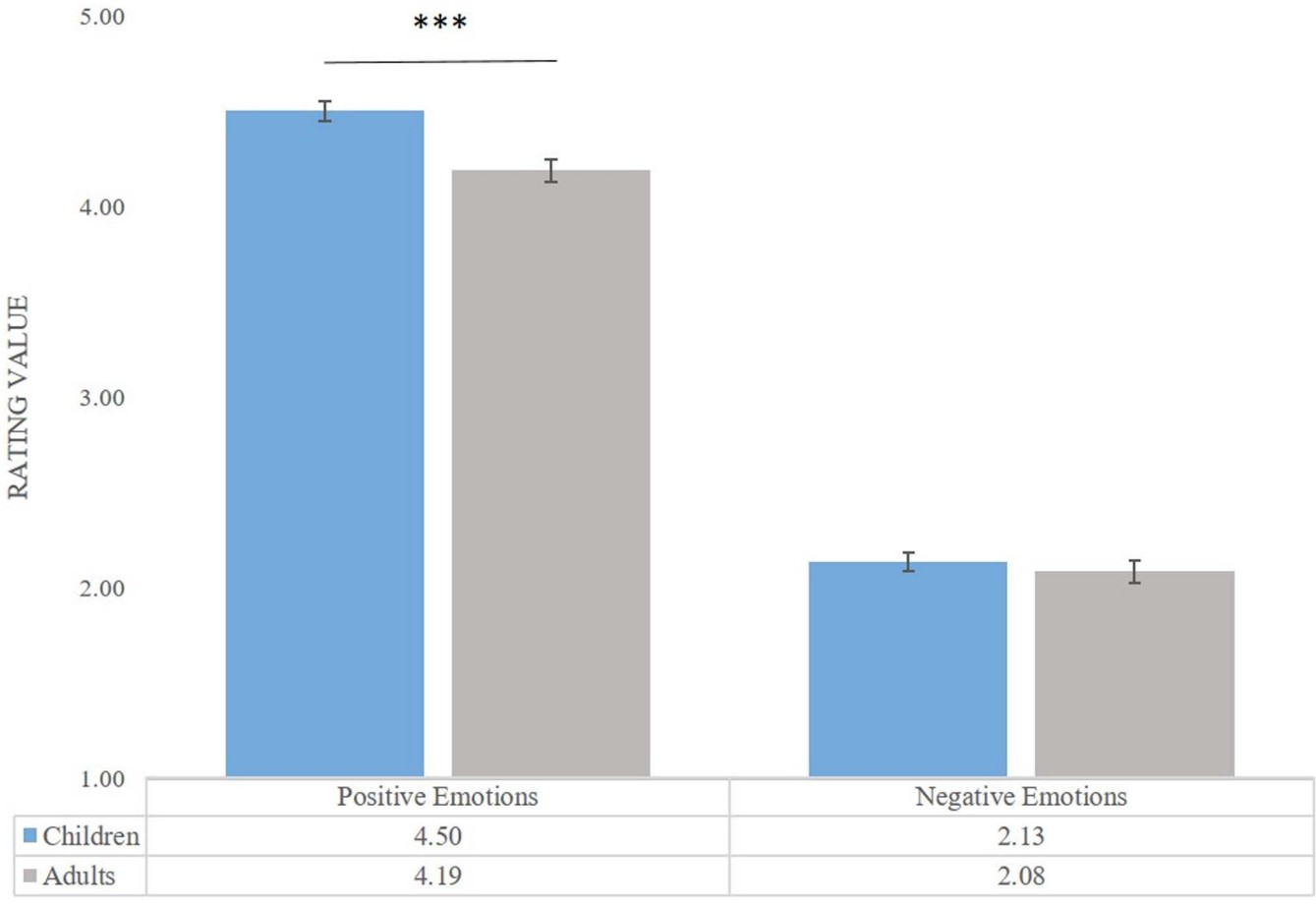

| | Positive Emotions | Negative Emotions |
|---|---|---|
| ■ Children | 4.50 | 2.13 |
| ■ Adults | 4.19 | 2.08 |

**Fig 7. Children's and adults' valence ratings.** Mean rating values are for positive and negative expressions across both stimulus versions (visual and visual-verbal). Error bars represent standard error. *p < .05 **p < .01. ***p < .001.

## Emotion-specific results

To better understand how specific emotions influenced participants' valence ratings, we conducted emotion-specific analyses within each age group. For children, we explored which positive emotion (happiness or happy-surprise) yielded higher positive ratings by conducting a paired samples $t$-test. The results showed that surprise and happiness were rated equally positive ($M_{surprise}$ = 4.52, $M_{happy}$ = 4.48, $p$ = .65). Additionally, we investigated which negative emotion (anger or sadness) was perceived as more negative by the adult sample. A paired samples $t$-test revealed that sadness was rated significantly lower in valence than anger ($M_{sad}$ = 1.82, $M_{anger}$ = 2.34, $p$ < .001).

Thus, while children did not differentiate between the two positive emotions, adults showed a clearer distinction between negative emotions, perceiving sadness as significantly more negative than anger.

## Observed power

We performed a post-hoc G-Power analysis (4 measurements, 56 participants) which revealed observed power of 93.1%, indicating ample power for this measure.

## Perception of stimuli: Uncanny Valley measures

In addition, we explored how participants from both age groups perceived the characters in terms of the Uncanny Valley (UV) effect. Using the responses of 14 children from the arousal sample and another 14 from the valence sample, we compared them to the responses of the full adult sample ($N$ = 28) for the three UV measures: Likeability, Friendliness, and Realism. Fig 8 illustrates the frequency of responses (in percentages). Overall, most children and adults rated the stimuli as likable and friendly. A notable difference between the groups emerged in the Realism scale: while more than half of the children rated the stimuli as low in realism, the majority of the adults considered them highly realistic. We performed three chi-square analyses to determine whether these frequency differences were significant, one for each UV measure.

For the Likeability scale, we found no significant difference between the groups, $X^2$ (2, $N$ = 56) = 1.37, $p$ = .50. Similarly, the Friendliness scale did not yield significant results, $X^2$ (2, $N$ = 56) = 5.14, $p$ = .07. The only significant difference was revealed for the Realism scale, $X^2$ (2, $N$ = 56) = 9.16, $p$ = .01, with children rating the stimuli as less realistic compared to the adults.

## Negativity bias and Uncanny Valley: correlations

Considering that more than half (61%) of the children perceived the stimuli as low in realism, we investigated whether this is connected to the negativity arousal bias found in their emotional ratings. We correlated children's rating values for the Realism scale with their ratings for the negative emotions in the arousal condition. No significant correlation was revealed,

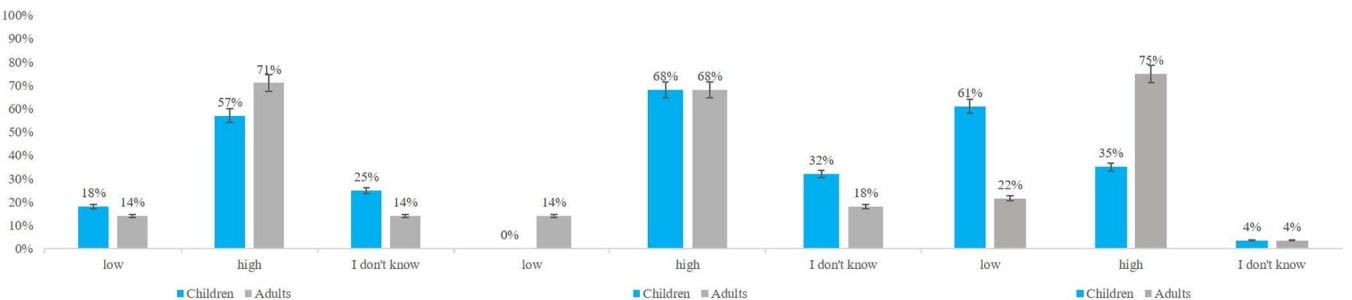

**Fig 8. Frequency of children's and adults' responses for the UV measures.** Left to right: likeability, friendliness and realism. The ratings are of the whole stimulus set across both valence category and stimulus types. The bar graphs show percentages. Error bars represent standard error.

$r_s(24) = -.008$, $p = .97$, indicating that their higher arousal ratings for negative emotions are not associated with the perceived low character realism. Additionally, we correlated children's ratings for the Realism scale and their ratings for positive emotions. Again, no significant correlation was found, $r_s(24) = -.004$, $p = .99$, suggesting no link between perceived realism and children's ratings of positive emotions.

## Discussion

Uncovering the mechanisms behind age-related changes in emotion perception from childhood to adulthood is essential for understanding the complexities of emotional development. While existing studies have contributed to this field using various emotional stimuli [14,17,18] and different presentation methods [42,43], there has been a notable lack of implementing naturalistic emotion stimuli.

The aim of this online rating study was to address this gap in the literature by exploring how children and adults perceive valence and arousal with a newly designed database that closely mirrors real-life emotion experiences. Our Meta-MED database features naturalistic emotion characteristics such as four dynamic, low-intensity emotional expressions (happiness, happy-surprise, sadness, anger) presented by faces of different age groups (children, adolescents, young adults, and older adults) in two modalities (pure facial expressions and combined facial-vocal expressions). It is crucial to note that the participants assessed the arousal and valence states of the digital avatars, rather than their subjective emotional reactions during the observation.

### Arousal ratings of children and adults

Our findings on arousal showed a strong similarity between children's and adults' ratings, as shown by the high correlation mirroring the results of Vesker et al. [14]. Despite this, there were significant developmental differences between the two age groups.

First, children exhibited a negativity bias, rating negative expressions as more arousing than adults. This heightened perception of arousal for negative emotions was consistent across both visual and visual-verbal conditions, suggesting a strong negative emotional component regardless of stimulus presentation. This contrasts with previous findings [14], which reported a positivity bias in children's arousal ratings. We suspect that this difference could stem from the dynamic nature of our stimuli. On the one hand, static stimuli showing emotions at their peak could lead to an almost immediate detection of a negative expression, thus potentially resulting in lower arousal ratings. On the other hand, dynamically unfolding expressions from neutral to negative might be perceived as unanticipated, and therefore more arousing upon reaching their peak. Such observations have been made by other researchers regarding static versus dynamic presentation of stimuli [70].

Additionally, we observed that anger, not sadness, elicited higher ratings in arousal, which is in line with previous research [11]. Based on this, we hypothesize one possible explanation for our findings. As discussed by Vaish [24] in their literature review, infants may attend to negative faces (such as anger or fear) more frequently than happy ones, given that these emotions signal potential danger, which is more relevant to the infant. From an evolutionarily perspective, we suspect that children, being less capable of defending themselves, could be more reactive to expressions of anger due to the perceived higher threat value. Interestingly, a similar framework was proposed by Vesker and team [14], where they found a positivity bias for arousal from static images. They suggest that children might seek positive information from their surrounding as a means of protection from threats. While their findings revealed a positivity bias, both studies align with the evolutionary mechanisms driving these biases, although the direction may vary based on the stimulus material. Considering that our stimuli, we suspect that the difference in the observed biases in our study and Vesker's study [14] could be attributed to the nature of our stimuli – dynamically unfolding and containing verbal information that might have enhanced the perception of emotions for both age groups due to the additional cues from the voice (e.g., an angry/sad tone).

Finally, we did not find consistent effects regarding arousal elicited by the characters' age groups. Older faces generally produced higher arousal ratings across all participants, yet this effect was not stable and varied across conditions. Nevertheless, incorporating emotional expressions from various facial ages is crucial for maintaining an ecological approach to studying emotion perception.

## Valence ratings of children and adults

One key highlight of the valence findings is the validation of our new stimulus database, demonstrated by the consistent and accurate valence categorization across all participants. This validation is crucial for its future applications, as it emphasizes the reliable construction of the emotional expressions.

Similar to the arousal ratings, the responses from both age groups were nearly identical across all stimuli. However, several significant differences emerged between and within the two age groups. Contrary to their negativity bias in arousal, children exhibited a positivity bias, with higher valence ratings for positive emotions in both stimulus conditions than adults. This aligns with two previous findings [14,16], where children rated and categorized positive expressions similarly in an experimental design featuring static facial expressions. This suggests a robust overall effect of positive emotions, regardless of the presentation type (visual/static or visual-verbal/bimodal). Regarding the emotions presented, children did not differentiate between surprise and happiness, which is expected considering we constructed both emotions as similar as possible (e.g., surprise ends with a smile) to be considered both positive, resulting in similar valence ratings.

A new discovery in the adult group showed a negativity bias in the visual condition compared to children. This effect was not observed in the visual-verbal condition, possibly because adults do not need additional cues to judge the valence of negative emotions. Interestingly, adults rated sadness as significantly more negative than anger, further aligning with Russell's model [11]. According to it, while anger has high arousal ratings and a rather higher valence score, sadness is a low-arousing emotion with a much lower (i.e., negative) valence score.

The results from both age groups suggest inherent differences in the processing of specific emotions in terms of valence. Since our happy and happy-surprise expressions were constructed similarly (i.e., with a smile), we did not expect significant differences just as our findings support. This would have probably been different had surprise been left ambiguous. However, the distinction in valence between anger and sadness, which we found in adults, was more pronounced, with sadness having a stronger negative perceptual impact. These findings align with the notion that "negative emotions are more richly differentiated than positive emotions" [71].

Finally, regarding the age variation of faces, an effect of the older faces was observed. Still, since it was limited to specific conditions and comparisons, drawing definitive conclusions regarding the influence of facial age on emotion perception is challenging.

## Effect of Metahumans on emotion perception

Another topic of interest was to determine whether the development and presentation of our stimuli had an impact on emotion perception. To determine this, we compared the two visually distinct character sets and asked our participants to rate their overall impression of the stimuli to measure potential Uncanny Valley effects [46].

We found that the two versions were rated similarly, showing that the MetaHuman character design did not affect emotion perception. This contributes to the strength of our findings by displaying the robustness of our stimuli construction. Furthermore, children and adults perceived the digital characters as pleasant and friendly, indicating an overall positive impression of the MetaHumans. The only notable difference observed was in perceived realism, with adults rating the avatars as more realistic than children did. This discrepancy may be related to differences in media exposure and realism familiarity. Adults are generally more accustomed to lifelike digital representations through experiences with video games, films, and other forms of digital media. In contrast, children are typically exposed to more stylized, caricatured animations,

which could influence their standard of realism. Supporting this idea, gaming statistics from 2018 [72] indicated that the average age of gamers was 34, with only 17% under the age of 18—suggesting that adults are more frequently engaged with realistic digital environments. Additionally, research has shown that adults tend to prefer realistic over stylized characters [73], whereas children often favor cartoon-like presenters in educational content [74].

## Future directions

The findings of the current online study not only validate our newly developed Meta-MED database but also highlight the importance of employing realistic and natural stimuli when examining age-related differences in emotion perception. There is growing consensus among researchers regarding the need for ecologically valid stimuli that closely resemble real-life experiences. Furthermore, drawing conclusions from studies using static stimuli should be taken with a grain of salt, as these may not accurately represent dynamic emotional expressions [75–77].

Our new database emphasizes the effectiveness of using highly realistic avatars to generate stimuli. These avatars are cost-effective, highly controllable, and offer flexibility in representing various aspects of emotional expression (such as intensity and duration) and identity (including race and gender). Recent advancements in MetaHuman technology, which enable real-time facial tracking, have significantly improved emotional expressiveness and can be accessed with just a smartphone tap. Needless to say, this is not easily replicated with hired actors, who often provide posed expressions that resemble static images, resulting in a similarly unrealistic emotion production.

Furthermore, an increasing number of studies have explored emotion perception using digital avatars, particularly in areas such as developing social and communication skills for autistic individuals [78] and examining empathy and facial mimicry [79,80], as well as comfort level and perceived realism of digital avatars [81]. These studies are paving the way for avatars to be used in various settings, including virtual reality (e.g., patient simulation: [82,83]), educational experiences [84,85], and interventions to enhance performance in individuals with Attention Deficit Hyperactivity Disorder [86–88].

In terms of emotional development, the differences observed in the child sample may have practical implications across various domains such as education, parenting, and overall cognitive development. For instance, educators could incorporate more positive reinforcement into their teaching methods to enhance students' understanding of educational content [89]. Interestingly, a case study has demonstrated that both positive and negative reinforcement can lead to improved academic performance and reduced problematic behaviors [90]. Additionally, this knowledge can be applied when developing support programs for youth navigating challenging and negative experiences. Such programs could focus on reframing these experiences with a more positive perspective, acknowledging consequences, fostering responsibility, and promoting growth [91].

Furthermore, parents and caregivers can benefit from creating a balanced emotional environment in which children learn to better perceive and regulate both their own and others' emotions. This can be achieved through encouragement, praise, and rewards [92].

Lastly, a deeper understanding of children's emotional development and needs could inform improvements in early childhood education and care practices. By reframing challenging and negative experiences in a positive light and encouraging responsibility, we can better support children's development and help them understand the role of these experiences in shaping their growth [91,93].

## Limitations

While we carefully planned and controlled all aspects of our experiment, there are several limitations. First, as an online study, we had no experimental control over how the stimuli were presented, meaning that the setting might have varied from one participant to another. This applies to the need for parental support, and while parents were instructed not to influence their children's rating decisions, we cannot be certain they complied. Still, considering that the developmental

differences we observed would not have emerged had this been the case, we regard this as an unlikely scenario. Second, for the construction of the stimuli, we chose a balanced presentation of emotional expressions – two positive and two negative – however, we acknowledge that nearly all research so far includes most, if not all, six basic emotions. Third, the sentences used for the visual-verbal condition were constructed to be simplistic; still, it is perhaps uncommon for people to express their feelings in such a straightforward manner in everyday life. Finally, while MetaHumans appear quite realistic, their complexity remains artificial, as reflected in the children's low realism responses. All these factors could have influenced how natural and real-life-like emotional expressions were perceived to be.

## Conclusion

The present online rating study featuring dynamic and multimodal emotion expressions revealed that children maintained their previously found positivity bias in valence even with dynamic stimuli, whereas adults exhibited a negativity bias. Contrary to previous findings, children rated negative expressions as more arousing. Furthermore, the type of presentation matters: multimodal presentation not only elevates arousal ratings but also enhances the valence perception of negative emotions, whereas unimodal presentation strengthens the positivity perceived in positive emotions. No consistent effects were detected regarding the ages of the characters' faces. Overall, our findings highlight the fundamental importance of using ecologically valid stimuli when exploring developmental differences in emotion perception.

## Supporting information

**S1_Trial example. The video with the specific stimulus is presented with either one of the two scales (for the children sample) or with both scales (for the adult sample).**
(TIF)

**S2_Correlational heatmaps. The figures show the correlations separately for the visual and the visual verbal condition of children' and adults' responses for arousal and valence. The legend explains how to read the trials properly.**
(PDF)

## Author contributions

**Conceptualization:** Nikol Tsenkova, Daniela Bahn, Christina Kauschke, Gudrun Schwarzer.

**Data curation:** Nikol Tsenkova.

**Formal analysis:** Nikol Tsenkova, Gudrun Schwarzer.

**Funding acquisition:** Christina Kauschke, Gudrun Schwarzer.

**Investigation:** Nikol Tsenkova.

**Methodology:** Nikol Tsenkova, Daniela Bahn, Christina Kauschke.

**Project administration:** Gudrun Schwarzer.

**Resources:** Gudrun Schwarzer.

**Software:** Nikol Tsenkova.

**Supervision:** Daniela Bahn, Christina Kauschke, Gudrun Schwarzer.

**Validation:** Daniela Bahn.

**Visualization:** Nikol Tsenkova.

**Writing – original draft:** Nikol Tsenkova.

**Writing – review & editing:** Nikol Tsenkova, Daniela Bahn, Gudrun Schwarzer.

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
