## [Decision Letter · Decision Letter 0]

26 Feb 2025

Dear Dr. Tsenkova,

Please be clear and detailed in addressing each of the reviewer's comments, which will undoubtedly help to improve the quality of the work..Be sure you are in compliance with the journal's guidelines and requirements.

We look forward to receiving your revised manuscript.

Kind regards,

Irving A. Cruz-Albarran

Academic Editor

PLOS ONE

Journal Requirements:

This work was supported by the DFG (Deutsche Forschungsgemeinschaft) [grant number SFB/TRR 540 135/3 2014].

5. Please remove your figures from within your manuscript file, leaving only the individual TIFF/EPS image files, uploaded separately. These will be automatically included in the reviewers’ PDF**.**

**6.** We note that Figure 1, Supporting figure includes an image of a [patient / participant / in the study].

7. Please remove all personal information, ensure that the data shared are in accordance with participant consent, and re-upload a fully anonymized data set.

Additional Editor Comments :

Reviewers' comments must be addressed in detail, in accordance with the journal's requirements and guidelines.

The graphs should be made using professional software. This will improve their quality.

Reviewers' comments:

Reviewer's Responses to Questions

**Comments to the Author**

1. Is the manuscript technically sound, and do the data support the conclusions?

Reviewer #1: Yes

Reviewer #2: No

Reviewer #3: Yes

2. Has the statistical analysis been performed appropriately and rigorously?

Reviewer #1: Yes

Reviewer #2: Yes

Reviewer #3: Yes

3. Have the authors made all data underlying the findings in their manuscript fully available?

Reviewer #1: Yes

Reviewer #2: Yes

Reviewer #3: Yes

4. Is the manuscript presented in an intelligible fashion and written in standard English?

Reviewer #1: Yes

Reviewer #2: No

Reviewer #3: Yes

Reviewer #1: SUMMARY

The authors present an insightful study on the life-course development of emotion processing through the unique combination of low intensity, dynamic, multi-modal, and age-variant stimuli. This approach offers greater ecological validity than is typically achieved in experiments on emotion recognition. The authors provide a strong rationale for why this unique combination – chosen to improve the ‘naturalness’ of the stimuli – contributes to an improved understanding of human emotion recognition across age-groups. The stimuli and method are rigorously described, aiding future replicability. Validation of the Meta-MED data set is a valuable contribution to the field and provides researchers with a vast new array of emotional stimuli with high ecological validity. However, I believe that the manuscript would benefit from further elaboration on the implications of these findings outside the validation of new stimuli. There are additionally some areas where the manuscript would benefit from some minor clarifications. I am grateful to the authors for their hard work on their original study – I wish them luck in their future endeavours.

STRENGTHS

1. It is positive to see that data has been made available through the OSF and I commend the authors’ transparency.

2. In a similar vein, I note that the authors provide access to their newly validated stimuli. As noted above, this is a valuable contribution to the field.

3. The authors have rigorously described their stimuli and methodology. This lends the study towards being highly replicable.

4. The authors have striven to ensure control over extraneous variables, such as the possible ‘uncanniness’ of the artificial stimuli, including suitable statistical investigations of these possible extraneous effects.

5. Results are well-reported and clear, aiding the reader’s understanding of a large set of statistical results.

MAJOR

- No major revisions identified

MINOR

1. An in-text citation for the ‘numerous studies’ referenced in line 55 would help direct readers to the most relevant studies and/or reviews.

2. Across lines 132-135, the authors explain that the valence of the bias induced during emotion recognition tasks (positive vs. negative) is task dependent. Given that the previous paragraphs detail age-dependent differences in positivity/negativity biases, it would be helpful if the authors could clarify whether of not these task-specific biases show any age-dependence or otherwise have the same effects upon children as adults.

3. Lines 169-170 describe how the dependent variables were split across groups for children but not adults. Including a line on the rationale for this split would aid clarity.

4. Captions for figures 4 and 5 would benefit from a note to define what asterisks represent (e.g., * = p < .05).

5. Missing close bracket on reference 32, line 571.

6. Regarding the evolutionary explanation across lines 578-581, where it is stated that children’s negativity bias is driven by heightened threat detections due to their relative vulnerability: Whilst I find this a plausible suggestion, in the absence of specifically testing this hypothesis – or indeed referencing work to support this notion – the authors may wish to consider the addition of an alternative, perhaps non-evolutionary, explanation for this finding. In its current form, I am concerned that the explanation is vulnerable to the ‘just so’ critique of evolutionary explanations in psychology.

Moreover, would the logic presented here not also dictate that angry expressions in physically imposing individuals (e.g., a male adult) should be perceived as more arousing to children than the angry expressions of less physically imposing individuals (e.g., a female child)? I note that the three-way interaction between Age of Faces, Valence Category, and Stimulus Type (described across lines 396-400) seemingly revealed no significant difference in young adults compared to other age groups for negative valence.

7. Further discussion of implications of this study would strengthen the manuscript. The authors do discuss the implications of their findings in relation to the validation of a new stimulus database – and indeed, this is a great strength of the study – however, currently lacking is further discussion of the findings’ real-world implications. For example, through lines 612 and 613, the authors report that ‘The results from both age groups suggest inherent differences in the processing of specific emotions in terms of valence.’ What are the broader implications of this knowledge on the ontogeny of emotion processing? The authors may wish to ground their findings in potential real-world applications (be that in educational settings, creating media, or parenting etc.)

Reviewer #2: The specific comments are as follows:

Abstract

Line 40-41, the authors said “our findings partially align with previous research…”, the statement “previous research” is not entirely appropriate in Abstract because it is too broad for readers. Specifically, it is difficult for readers to have a clear understanding of the term “previous research” when it first appeared in Abstract section.

Keywords

The word “development” is not suitable as a keyword, and it is recommended to select a suitable one again.

Introduction

The structure and presentation of this section are not clear. It is recommended to reorganize the new structure based on the research topic and content. Overall, the introduction section needs a clear main line to gradually induce the purpose, significance, and innovation of the current research.

Specifically,

Line 51-61, this section is too short and there is too little preparation in this section. It is recommended to provide necessary research background and add necessary connecting paragraphs to continue the context.

It is necessary to elaborate on the research progress related to emotion perception in detail, in order to reasonably introduce the innovation of current study.

It is necessary to clearly indicate the new contributions and values of current research in this field or related research areas.

Line 63 Naturalness of emotional stimuli in prior studies

Line 107 Development of emotion perception

Line 137 Usage of Metahumans

Do the above three parts belong to the literature review? If so, please provide a clear structural explanation, for example adding a new section literature review. If not, the necessary explanations should also be provided to facilitate readers' better understanding of the authors' work. In addition, for each individual part, the authors should provide necessary summaries rather than just comments.

Methods

Line 168 Participants, in this study, the number of participants in each group seems to be very small. Can this small sample size meet the requirements of data statistics? How to ensure the reliability and applicability of data results?

Line 266-267, the authors said “three additional scales”, Have the authors conducted a reliability and validity test of these scales?

Results

Line 306-307, although the authors stated that gender differences are not significant, it is best to present data results to support it. In fact, gender is a good factor worth exploring the difference between them.

Line 314, The writing of “t” in the expression of “t-tests” should be italicized, namely, “t-tests”

Line 357, the authors were advised to try changing the presentation format of Figure 2, 3. 5, and 6, such as replacing the current wireframe with a correlation heatmap (including correlation coefficients).

Discussion

Line 572-576, the authors' discussion are suggested to cite necessary references to support the analysis, thereby enhancing the reliability of the discussion.

Line 601, “aligns with previous findings [32]”, the cited reference is the same as those in line 571. It is recommended that the authors search for new relevant literature to support discussion and avoid repeatedly citing the same article.

Line 602-604, the authors' discussion are suggested to cite necessary references to support the analysis, thereby enhancing the reliability of the discussion.

Line 634-637, the authors' discussion are suggested to cite necessary references to support the analysis, thereby enhancing the reliability of the discussion.

Conclusions

Line 660, “likely due to”, it is not appropriate to use the term 'due to' in the conclusion section, because it is widely used in discussion section. The conclusion section should be clear research findings obtained, rather than uncertain or ambiguous statements.

References

Some of the cited references are too outdated. It is recommended to update or supplement with new references.

Figures

The presentation of graphics should be improved to enhance their aesthetic appeal and clarity.

Reviewer #3: The study addresses an extremely relevant and current topic, contributing to its potential impact in the research field. The selected bibliography is well-chosen and up-to-date, providing a solid theoretical foundation and reinforcing the study's scientific grounding.

Despite the significant interest and value of the article, the introduction presents opportunities for improvement, particularly in terms of its organization. Better structuring of this section could lead to a more fluid and coherent reading experience, allowing readers to quickly grasp the research problem, study objectives, and relevance within the field. The introduction could include a clearer exposition of the research gap this study aims to fill, along with a more structured argumentation regarding its importance and innovation, always grounded in previous similar studies.

The methodological description could be more detailed, particularly regarding the participant recruitment process and the type of sampling used. These details are crucial to ensuring research transparency, allowing other researchers to understand and potentially replicate the study. A more precise and rigorous description of the employed method would strengthen the credibility of the results and help justify the methodological choices made throughout the research.

Regarding data presentation, the graphs used are well-constructed and visually clear, facilitating the understanding of the results. However, it would be beneficial if these representations were accompanied by a more in-depth analysis, highlighting their relevance within the research context and providing detailed interpretations of what the data reveal. The inclusion of comparisons with other studies or more extensive explanations could further enrich this section.

Another point that deserves discussion is the designation of so-called "negative emotions." While this term is widely used in the literature, it would be worth questioning its appropriateness. Is the categorization of emotions as "negative" the most suitable? Is there a solid theoretical foundation that justifies this terminology? If not, adopting an alternative nomenclature (such as "unpleasant emotions") or at least discussing the implications of using this concept in the study’s field could be more productive.

Finally, an essential aspect to address is the comparability of this study with previous research. To what extent do the innovations brought by this work allow direct comparisons with other published studies? The study could benefit from a deeper analysis of how it positions itself within the existing scientific landscape, highlighting similarities and differences concerning prior research. This type of discussion could help contextualize the findings more effectively and reinforce the originality of the research.

Overall, the study presents clear and significant strengths. However, with some improvements, it could achieve an even higher level of clarity, scientific rigor, and contribution to the field.

**Do you want your identity to be public for this peer review?** For information about this choice, including consent withdrawal, please see our Privacy Policy

Reviewer #1: No

Reviewer #2: No

Reviewer #3: No

---

## [Author Response · Author response to Decision Letter 1]

12 Apr 2025

Dear Mr. Cruz-Albarran,

Thank you very much for evaluating our manuscript and giving us the opportunity to revise it. Attached please find the revised version of the manuscript and our detailed replies to each of the concerns raised by the reviewers.

We have carefully responded to all of the suggestions and, for example, restructured the Introduction for better flow and coherence as suggested by the reviewers, and added a Future Directions subheading in the Discussion to address the applications of our findings.

We would like to thank you and all three reviewers for the helpful comments and the overall positive feedback.

We hope that our revision satisfactorily addresses all concerns and look forward to your decision.

Sincerely,

Nikol Tsenkova

Replies to Reviewer 1

1. An in-text citation for the ‘numerous studies’ referenced in line 55 would help direct readers to the most relevant studies and/or reviews.

Thank you for this observation. In the revised manuscript, we have added additional references to support the statement in line 55 (original), which now appear in lines 54-56.

“Consequently, numerous studies have focused on exploring the developmental trajectory of emotion perception across various age groups – mostly in children [5-6] and adults [7], but also in adolescents [8-9], and older adults [10].”

2. Across lines 132-135, the authors explain that the valence of the bias induced during emotion recognition tasks (positive vs. negative) is task dependent. Given that the previous paragraphs detail age-dependent differences in positivity/negativity biases, it would be helpful if the authors could clarify whether of not these task-specific biases show any age-dependence or otherwise have the same effects upon children as adults.

Thank you for this insightful comment. We have clarified whether these findings discussed in lines 132-135 (original) are in children, adults or both age groups (now in lines 81-85).

While some tasks are more likely to produce a positivity bias (e.g., identification tasks: children [19], adults [20]; intensity and arousal ratings: children [21]), others have shown to induce a negativity bias (e.g., visual search tasks: both children and adults [22]; recognition tasks: younger and older adults [23]; for a comprehensive review on infants and children, see [24, 25]).

3. Lines 169-170 describe how the dependent variables were split across groups for children but not adults. Including a line on the rationale for this split would aid clarity.

We fully agree with the reviewer's suggestion and have now added an additional explanation for our decision to split the children groups (lines 154-158 in revised manuscript).

“We tested two separate groups of German children, one for each of the two dependent measures (arousal and valence), and one group of adults who performed both measures simultaneously. In order to avoid fatigue and confusion regarding the difference between the scales, we separated the children sample into two groups. We did not expect such issues with the adult sample, hence there was only one adult group.”

4. Captions for figures 4 and 5 would benefit from a note to define what asterisks represent (e.g., * = p < .05).

Thank you, we have now added that.

5. Missing close bracket on reference 32, line 571.

Thank you for bringing this to our attention. It is now fixed!

6. Regarding the evolutionary explanation across lines 578-581, where it is stated that children’s negativity bias is driven by heightened threat detections due to their relative vulnerability: Whilst I find this a plausible suggestion, in the absence of specifically testing this hypothesis – or indeed referencing work to support this notion – the authors may wish to consider the addition of an alternative, perhaps non-evolutionary, explanation for this finding. In its current form, I am concerned that the explanation is vulnerable to the ‘just so’ critique of evolutionary explanations in psychology.

Thank you for highlighting this crucial point. We have addressed this issue by citing several works that support our observations and have improved the paragraph between lines 573-588.

“Additionally, we observed that anger, not sadness, elicited higher ratings in arousal, which is in line with previous research [11]. Based on this, we hypothesize one possible explanation for our findings. As discussed by Vaish [24] in their literature review, infants may attend to negative faces (such as anger or fear) more frequently than happy ones, given that these emotions signal potential danger, which is more relevant to the infant. From an evolutionarily perspective, we suspect that children, being less capable of defending themselves, could be more reactive to expressions of anger due to the perceived higher threat value. Interestingly, a similar framework was proposed by Vesker and team [14], where they found a positivity bias for arousal from static images. They suggest that children might seek positive information from their surrounding as a means of protection from threats. While their findings revealed a positivity bias, both studies align with the evolutionary mechanisms driving these biases, although the direction may vary based on the stimulus material. Considering that our stimuli, we suspect that the difference in the observed biases in our study and Vesker’s study [14] could be attributed to the nature of our stimuli – dynamically unfolding and containing verbal information that might have enhanced the perception of emotions for both age groups due to the additional cues from the voice (e.g. an angry/sad tone).”

Moreover, would the logic presented here not also dictate that angry expressions in physically imposing individuals (e.g., a male adult) should be perceived as more arousing to children than the angry expressions of less physically imposing individuals (e.g., a female child)? I note that the three-way interaction between Age of Faces, Valence Category, and Stimulus Type (described across lines 396-400) seemingly revealed no significant difference in young adults compared to other age groups for negative valence.

Regarding the comment on suspected higher arousal for angry adult faces compared to angry children faces, we decided to run an additional analysis with the few trials featuring angry faces. We performed a paired-samples t-test to determine whether children participants found adult angry faces as more arousing compared to children’s angry faces. However, we found no significant difference, t(25) = -0.62, p = .53. It is important to note that we only had four trials for each group (adult angry faces and child angry faces), which may have limited the statistical power of this analysis. With a larger number of trials, it could be possible to better explore this potential effect.

7. Further discussion of implications of this study would strengthen the manuscript. The authors do discuss the implications of their findings in relation to the validation of a new stimulus database – and indeed, this is a great strength of the study – however, currently lacking is further discussion of the findings’ real-world implications. For example, through lines 612 and 613, the authors report that ‘The results from both age groups suggest inherent differences in the processing of specific emotions in terms of valence.’ What are the broader implications of this knowledge on the ontogeny of emotion processing? The authors may wish to ground their findings in potential real-world applications (be that in educational settings, creating media, or parenting etc.)

We are extremely grateful for this suggestion and have made the necessary changes by introducing a new subheading within Discussion, entitled “Future Directions” (lines 651-690). We first discuss the importance of employing ecologically valid stimuli for exploring developmental differences in emotion perception (lines 651-657). Then we address the effectiveness of using digital avatars for such endeavors (lines 658-665). Furthermore, we highlight several studies that have successfully implemented digital avatars in areas like teaching people with autism social skills, aiding people with Attention Deficit Hyperactivity Disorder with their overall performance, as well as serving well in educational settings (lines 666-672). Lastly, and most importantly, we underscore the application of our findings in terms of teaching and parenting styles implementing positive reinforcement, as well as the importance of reframing negative experiences to crucial for developing important life skills in youth programs and in early childhood education and care practices (lines 673-690).

Replies to Reviewer 2

Abstract

Line 40-41, the authors said “our findings partially align with previous research…”, the statement “previous research” is not entirely appropriate in Abstract because it is too broad for readers. Specifically, it is difficult for readers to have a clear understanding of the term “previous research” when it first appeared in Abstract section.

Thank you for your suggestion, we have now removed that!

Keywords

The word “development” is not suitable as a keyword, and it is recommended to select a suitable one again.

We acknowledge that the keyword “development” is too general and thus have changed it to “emotional development” to be more specific and in line with the manuscript’s topic.

Introduction

The structure and presentation of this section are not clear. It is recommended to reorganize the new structure based on the research topic and content. Overall, the introduction section needs a clear main line to gradually induce the purpose, significance, and innovation of the current research.

Specifically, Line 51-61, this section is too short and there is too little preparation in this section. It is recommended to provide necessary research background and add necessary connecting paragraphs to continue the context.

It is necessary to elaborate on the research progress related to emotion perception in detail, in order to reasonably introduce the innovation of current study.

It is necessary to clearly indicate the new contributions and values of current research in this field or related research areas.

Line 63 Naturalness of emotional stimuli in prior studies

Line 107 Development of emotion perception

Line 137 Usage of Metahumans

Do the above three parts belong to the literature review? If so, please provide a clear structural explanation, for example adding a new section literature review. If not, the necessary explanations should also be provided to facilitate readers' better understanding of the authors' work. In addition, for each individual part, the authors should provide necessary summaries rather than just comments.

We completely agree with the reviewer’s observation and are grateful for the suggestions. To address this, we have restructured the introduction in several ways.

For clarity, we have removed the subheadings and instead have focused on bridging the separate parts of the introduction with summaries and connecting sentences. Additionally, we have rearranged certain parts. We first start with an overview of developmental changes in emotional development (line 51 to 70), report observed biases in the literature on developmental emotion perception (line 70 to 85), underscore reasons for the issue of inconsistent bias findings and report studies that have addressed these issues (line 86 to 122), and finally, propose a new solution for exploring age-related differences in emotion perception, i.e. by employing digital avatars (line 123 to 141). We conclude with the aims of this study (line 142 to 148).

Methods

Line 168 Participants, in this study, the number of participants in each group seems to be very small. Can this small sample size meet the requirements of data statistics? How to ensure the reliability and applicability of data results?

Thank you for suggesting this clarification. We have mentioned our a priori power analysis in lines 175 and 180, where we calculated that a total of 36 participants would be enough to test our hypothesis. We have collected data from a total of 82 participants (28 adults for both measures, 26 children for arousal, and 28 children for valence). Additionally, we have done post hoc power analyses found in lines 418-420 (for arousal):

“We performed a post-hoc G-Power analysis for arousal (8 measurements, 54 participants) revealed observed power of 71.9%, slightly below the conventional threshold of .80.”

And in lines 507-509 (for valence):

“We performed a post-hoc G-Power analysis (4 measurements, 56 participants) which revealed observed power of 93.1%, indicating ample power for this measure.”

Line 266-267, the authors said “three additional scales”, Have the authors conducted a reliability and validity test of these scales?

Thank you for bringing this to our attention. We have now included a thorough explanation of the scales and several citations showing the validity and reliability of the mentioned scales (line 264-271):

“The Godspeed Questionnaire (GQ) is the most frequently used tool in the field of Human-Robot Interaction [65], with over 3,000 citations as of April 2025, and has been translated into 19 languages. It consists of five scales, which can be used independently: Anthropomorphism (α = 0.87), Animacy (α = 0.92), Likeability (α = 0.70), Perceived Intelligence (α = 0.75), and Perceived Safety (α = 0.91) [66]. The German version of the GQ [67] has been reported to have good internal reliability (α = 0.70). In terms of validity, only the Polish version [68] has undergone factor analysis, which yielded a total variance of 74.24% for the four-factor solution.”

Results

Line 306-307, although the authors stated that gender differences are not significant, it is best to present data results to support it. In fact, gender is a good factor worth exploring the difference between them.

Thank you for this comment. We have now added this information in lines 311-314.

“Furthermore, we found no significant gender effects for either of the two measures – for arousal, F(1, 50) = .11, p = .73, η2p = .002, for valence, F(1, 52) = .23, p = .63, η2p = .004. Thus, we did not include gender in all further analyses.”

Line 314, The writing of “t” in the expression of “t-tests” should be italicized, namely, “t-tests”

Thank you, we have now changed this.

Line 357, the authors were advised to try changing the presentation format of Figure 2, 3. 5, and 6, such as replacing the current wireframe with a correlation heatmap (including correlation coefficients).

Thank you for your insightful suggestion! We have added four correlational heatmaps as Supplementary Material, showing the specific correlational coefficient of the responses between children and adults for each stimulus. We could not include them in the manuscript due to the large size of the correlational heatmap and considered it would be best to have this information in Supplementary Material instead.

Discussion

Line 572-576, the authors' discussion are suggested to cite necessary references to support the analysis, thereby enhancing the reliability of the discussion.

Thank you for your suggestion; we have now added a reference to support our observations (lines 569-572).

Line 601, “aligns with previous findings [32]”, the cited reference is the same as those in line 571. It is recommended that the authors search for new relevant literature to support discussion and avoid repeatedly citing the same article.

Thank you for this suggestion. We acknowledge the repeated citation and have now included an additional study by the same author, which - although based on a different task - shares important similarities with the design and goals of our own experiment. It was challenging to find further relevant literature, considering the limited number of studies that specifically examine arousal and valence ratings across both children and adults, particularly in a non-correlational context. Considering our study’s design and research questions which are closely aligned with the original paper cited, and given that it is, to our knowledge, the only one addressing these specific questions, we focused our comparis

---

## [Decision Letter · Decision Letter 1]

3 Jul 2025

Dear Dr. Tsenkova,

Thank you for submitting your manuscript to PLOS ONE. After careful consideration, we feel that it has merit but does not fully meet PLOS ONE’s publication criteria as it currently stands. Therefore, we invite you to submit a revised version of the manuscript that addresses the points raised during the review process.

We look forward to receiving your revised manuscript.

Kind regards,

Irving A. Cruz-Albarran

Academic Editor

PLOS ONE

Journal Requirements:

Additional Editor Comments :

Thank you very much for addressing the reviewers' comments correctly. We kindly ask you to address the minor revisions requested by the last reviewer. Thank you again.

Reviewers' comments:

Reviewer's Responses to Questions

**Comments to the Author**

Reviewer #1: All comments have been addressed

Reviewer #4: All comments have been addressed

2. Is the manuscript technically sound, and do the data support the conclusions?

Reviewer #1: Yes

Reviewer #4: Yes

3. Has the statistical analysis been performed appropriately and rigorously?

Reviewer #1: Yes

Reviewer #4: Yes

4. Have the authors made all data underlying the findings in their manuscript fully available?

Reviewer #1: Yes

Reviewer #4: Yes

5. Is the manuscript presented in an intelligible fashion and written in standard English?

Reviewer #1: Yes

Reviewer #4: Yes

Reviewer #1: The authors have adequately addressed all my previous comments. I extend my thanks to them for their efforts and wish them the best of luck in their future research.

Reviewer #4: I was not one of the original reviewers, and I can see that their comments have been thoroughly addressed, so my comments refer to other small pieces within the text.

My first comment refers to consent; you state that parents gave consent for their children, but were children asked if they wanted to take part? At that age they would have been able to understand and give assent. Similarly, were children and young people able to give consent for their voices to be used for the vocalisations, or did parents give this consent too?

My second comments is around payment, why did child participants get paid less than the adults?

Thirdly, I was wondering if the race of the avatars made any difference to the perceptions by participants. I know there is some evidence people are better at recognising emotions from those with a similar ethnic background, was this analysis carried out/possible?

Finally, autistic communities prefer 'autistic' rather than 'with autism', using this language would bring the terminology up to date and in line with the autistic people's preferences.

Overall this is a great article, and will add to the field. I found the uncanny valley effect analysis particularly interesting and look forward to seeing it published.

**Do you want your identity to be public for this peer review?** For information about this choice, including consent withdrawal, please see our Privacy Policy

Reviewer #1: No

Reviewer #4: No

---

## [Author Response · Author response to Decision Letter 2]

9 Jul 2025

Response to Reviewer #4:

1. My first comment refers to consent; you state that parents gave consent for their children, but were children asked if they wanted to take part? At that age they would have been able to understand and give assent. Similarly, were children and young people able to give consent for their voices to be used for the vocalisations, or did parents give this consent too?

- Thank you for this important comment. Since the study was conducted online, parents were instructed to involve their children only if they were willing and comfortable taking part. In two cases, children declined participation, which we fully respected.

Regarding the voice recordings, parents again obtained verbal assent from their children, who were informed about the purpose and nature of the recordings. All recordings were made with the children's full knowledge and agreement.

2. My second comments is around payment, why did child participants get paid less than the adults?

- Thank you for the question. The payment difference reflects the shorter duration of the task for children. They rated only half the videos using one scale (about 30 minutes total), whereas adults completed both scales across the full set of 128 videos, which took significantly longer.

3. Thirdly, I was wondering if the race of the avatars made any difference to the perceptions by participants. I know there is some evidence people are better at recognising emotions from those with a similar ethnic background, was this analysis carried out/possible?

- Thank you for this insightful comment. We considered this aspect during the creation of our database, but as the current study did not aim to examine race-related effects, we chose to use only Caucasian avatars to maintain consistency and reduce variability. As such, no analysis on racial bias was possible. However, we plan to expand the database to include more diverse facial representations in future work.

4. Finally, autistic communities prefer 'autistic' rather than 'with autism', using this language would bring the terminology up to date and in line with the autistic people's preferences.

- This is a very helpful comment, it’s much appreciated! We have now changed it as you have suggested.

We would like to thank the reviewer for all the valuable comments and suggestions that improved our manuscript.

---

## [Editor Report · Decision Letter 2]

18 Jul 2025

Developmental differences in perceiving arousal and valence from dynamically unfolding emotional expressions

PONE-D-24-37652R2

Dear Dr. Tsenkova,

We’re pleased to inform you that your manuscript has been judged scientifically suitable for publication and will be formally accepted for publication once it meets all outstanding technical requirements.

Kind regards,

Irving A. Cruz-Albarran

Academic Editor

PLOS ONE

Additional Editor Comments (optional):

Thank you very much for your attention to the comments.
---

## [Editor Report · Acceptance letter]

PONE-D-24-37652R2

PLOS ONE

Dear Dr. Tsenkova,

I'm pleased to inform you that your manuscript has been deemed suitable for publication in PLOS ONE. Congratulations! Your manuscript is now being handed over to our production team.

Kind regards,

on behalf of

Dr. Irving A. Cruz-Albarran

Academic Editor

PLOS ONE